# BoneMet: An Open Large-Scale Multi-Modal Murine Dataset for Breast Cancer Bone Metastasis Diagnosis and Prognosis

**Tiankuo Chu**[1], **Fudong Lin**[2], **Shubo Wang**[1], **Jason Jiang**[3], **Wiley J.W. Gong**[1],
**Xu Yuan**[2*], **Liyun Wang**[1†]

[1]Department of Mechanical Engineering, University of Delaware
[2]Department of Computer & Information Sciences, University of Delaware
[3]Department of Computer Science, University of Maryland

## Abstract

Breast cancer bone metastasis (BCBM) affects women's health globally, calling for the development of effective diagnosis and prognosis solutions. While deep learning has exhibited impressive capacities across various healthcare domains, its applicability in BCBM diseases is consistently hindered by the lack of an open, large-scale, deep learning-ready dataset. As such, we introduce the Bone Metastasis (BoneMet) dataset, the first large-scale, publicly available, high-resolution medical resource, which is derived from a well-accepted murine BCBM model. The *unique advantage* of BoneMet over existing human datasets is repeated sequential scans per subject over the entire disease development phases. The dataset consists of over *67 terabytes* of multi-modal medical data, including 2D X-ray images, 3D CT scans, and detailed biological data (*e.g.*, medical records and bone quantitative analysis), collected from more than five hundreds mice spanning from 2019 to 2024. Our BoneMet dataset is well-organized into six components, i.e., Rotation-X-Ray, Recon-CT, Seg-CT, Regist-CT, RoI-CT, and MiceMediRec. We further show that BoneMet can be readily adopted to build versatile, large-scale AI models for managing BCBM diseases in terms of diagnosis using 2D or 3D images, prognosis of bone deterioration, and sparse-angle 3D reconstruction for safe long-term disease monitoring. Our preliminary results demonstrate that BoneMet has the potentials to jump-start the development and fine-tuning of AI-driven solutions prior to their applications to human patients. To facilitate its easy access and wide dissemination, we have created the BoneMet package, providing three APIs that enable researchers to (i) flexibly process and download the BoneMet data filtered by specific time frames; and (ii) develop and train large-scale AI models for precise BCBM diagnosis and prognosis. The BoneMet dataset is officially available on Hugging Face Datasets at https://huggingface.co/datasets/BoneMet/BoneMet. The BoneMet package is available on the Python Package Index (PyPI) at https://pypi.org/project/BoneMet. Code and tutorials are available at https://github.com/Tiankuo528/BoneMet.

## 1 Introduction

Breast cancer stands as one of the most prevalent non-skin cancers affecting women globally (6). Although the survival rate for breast cancer patients has been greatly improved due to the adoption of cancer screening, early diagnosis, and more effective cancer treatments, the metastasis of breast cancer to other organs including bone dramatically reduces the five-year survival rate and worsens the patient suffering including severe pain, impaired mobility, and elevated risk of fatal fractures (11). Once invading the bone, breast cancers activate bone resorption, which in turn drives cancer growth. The "vicious cycle" between tumor and bone leads to the rapid decline of bone structural integrity (12). The

---

*Corresponding author: Dr. Xu Yuan (xyuan@udel.edu)

†Corresponding author: Dr. Liyun Wang (lywang@udel.edu)

conventional approaches for managing Breast cancer bone metastasis (BCBM) diseases utilize various medical imaging modalities including i) 2D X-ray images (13; 30), ii) 3D computed tomography (CT) scans (32; 3), and iii) combined localized metabolic imaging and CT scans (8; 46). However, these imaging based diagnosis and prognosis approaches suffer from several limitations, such as low sensitivity to small lesions in early-stage bone metastases (10), health risks of CT due to the exposure to high ionizing radiation (5), and interpretation variability limited by radiologist's experience and expertise (33). The lack of sensitivity and specificity of the imaging tools and algorithms leads to the current clinical guidelines against active monitoring of BCBM because the harms of false positive diagnosis and radiation from longitudinal imaging outweigh potential benefits (15). This poses a classic catch-22 dilemma. With clinicians being hesitant to image, we cannot feed and train AI models for better diagnosis. Conversely, without effective early detection tools, we miss the critical window of treating BCBM and fail to improve patient outcomes in a meaningful way.

Deep learning-based methods have shown significant improvements in identifying subtle patterns and features that are likely missed by human eyes, in particularly when analyzing large volumes of longitudinal data, reducing the scanning duration (radiation exposure) with efficient 3D CT reconstruction, and providing consistent and objective analysis and more comprehensive insights into patient health (24; 23; 44; 36). For BCBM diagnosis and prognosis applications, previous studies (29; 47; 18) have attempted to leverage deep learning models, while their performances appear to be hindered with low resolution images, inadequate diagnosis/prognosis labels (17), limited dataset sizes (1; 48; 14), and lack of diverse modalities (20). Hence, although the utilization of AI models to manage BCBM diseases offers a promising direction, the scarcity of large-scale, high-quality, relevant datasets—containing high-resolution X-ray and CT images along with detailed medical records—greatly impedes the development and application of large-scale AI models.

To further unleash the power of deep learning in BCBM diagnosis and prognosis, we first attack the major obstacle—the scarcity of high-resolution image datasets of early-stage BCBM with sufficient diagnostic and prognostic labeling—with our open-sourced large-scale BoneMet datasets. We leverage the large preclinical images acquired over the past five years (i.e., 2019-2024) from well-established mouse BCBM models (39). Mouse models, despite limitations, are valuable tools in elucidating disease mechanisms, identifying diagnostic biomarkers, and testing treatments (9). The small-sized mouse skeleton allows quick three-dimensional CT scans (< 4 min per scan) at high resolution (7-10 micron per pixel) and low radiation exposure (<0.4 Gy) for multiple time points over the span of breast cancer metastasis (39). Additional benefits include i) the complete control/documentation of the age of the experimental mice, the bone site of cancer cell metastasis, and the type and number of breast cancer cells being introduced in each mouse, ii) the application of various treatments to mimic human patient conditions such as chemo and radiation therapy, as well as exercise regimen (40). The rigorous experimental design contains age-matched non-tumor and placebo-treated controls, and thus the acquired image dataset reflects the complexity of human breast cancer progression with superior image quality and labeling over available patient data. We further demonstrate the utility of our BoneMet datasets in supporting the development of imaging biomarkers and AI applications, such as multimodal large vision models (MLVM) with temporal and spatial alignment, for the early diagnosis and prognosis of breast cancer bone metastasis as well as improved predictive accuracy of the models used in clinical settings.

Specifically, our dataset is collected from more than five hundreds mice spanning 5 years from 2019 to 2024, with its total size over *67 terabytes*. Each mouse undergoes 4 to 5 weekly sequential skeletal scans, including scans of the tibiae, distal femurs and some vertebrae. Each session produces 260 high-resolution, multi-angle 2D X-ray images at 0.8° intervals. From these X-ray images, 3D reconstructed CT images, segmented and registered CT scans, and RoI-cropped CT images of the tibiae are obtained using processing tools such as commercial software and our developed APIs. In addition, detailed biological data is recorded for each mouse, capturing critical medical details such as age, body weight, sex, metastatic tumor growth status, bone structural and mechanical properties, and other relevant data at each time point. To facilitate its effective use, this dataset is well-organized into six key components: Rotational X-Ray Imagery (**Rotation-X-Ray**), Reconstructed CT Imagery (**Recon-CT**), Segmented CT Imagery (**Seg-CT**), Registered CT Imagery (**Regist-CT**), Region of Interest CT Imagery (**RoI-CT**), and Mice Medical Records (**MiceMediRec**). Figure 1 provides an overview of our BoneMet dataset, showcasing examples of its six components, the processing tools used to acquire them, and the detailed data collection procedures. Besides, we have developed the BoneMet package, which includes three types of APIs for CT image segmentation, CT image

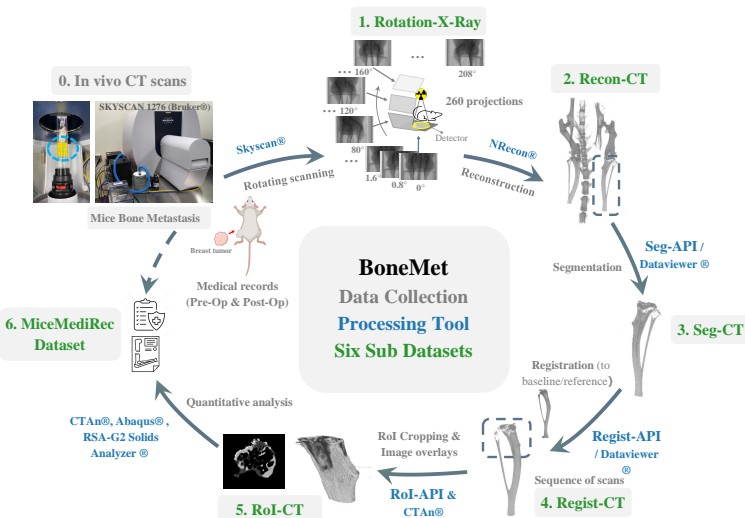

Figure 1: The data collection, processing tools, and six components for our BoneMet dataset.

registration, and RoI-based CT image cropping, respectively. These tools have been released on the Python Package Index (PyPI), aimed at assisting researchers and practitioners by providing efficient, standardized methods for data processing. These APIs enhance reproducibility, streamline the processing of large datasets, offer user-friendly interfaces, and ensure consistent data handling.

To the best of our knowledge, this dataset is the first of its kind, offering large-scale, high-resolution, and multi-modal medical data specifically targeting the management of BCBM diseases. It can support both supervised BCBM diagnosis and prognosis, thanks to its ground-truth labels, and self-supervised pre-training, due to its vast amount of data samples. Our experimental results validate the comprehensive applicability of our BoneMet dataset.

## 2 OUR BONEMET DATASET

Our BoneMet dataset is an open, large-scale, and multi-modal resource specifically designed for BCBM diseases. It offers high-resolution multi-modal medical data, including 2D X-ray images, 3D CT scans, and medical records and quantitative analysis. This section details the collection and preparation process of BoneMet dataset, its contents, and potential applications.

### 2.1 DATA COLLECTION AND PREPARATION

This dataset spans five years, from 2019 to 2024, and includes data over 500 either C57BL/6J or Balb/C mice with the age varying from young (8 weeks) to aged (>70 weeks) mice. These mice were divided randomly assigned into non-tumor and tumor groups, which further received individual or combined cancer interventions such as exercise, radiation therapy, chemotherapy and other experimental drugs (39). Breast cancer cells (1000-20,000 Py8119 cells or 3000-5000 4T1 cells) were directly injected into the proximal end of the mouse tibia of the tumor groups. Some mice received tumor injections on both tibiae, while some mice received injection on one tibia. The tumor and non-tumor carrying mice undergoing interventions or placebo treatments were then longitudinally monitored by microCT scans over a period of 3-5 weeks. The overview of data collection and preparation procedures is illustrated in Figure 1, and the details are presented as below, which produce the BoneMet dataset into six well-organized components.

*First,* the **Rotational X-Ray Imagery (Rotation-X-Ray)** is captured weekly by the SkyScan® 1276 scanner (Bruker, Kontich, Belgium), starting from the beginning of the breast tumor inoculation to the end of the sacrifice of the mice for 3 to 5 weeks. The camera revolved around the anesthetized animal's body from the anterior (front view) to the lateral and posterior (back view) aspects of the hind limbs with a step of 0.8 degrees, with 260 X-ray projections for each scan and covering a total rotation of 208°. The Rotation-X-Ray dataset collected in each week are organized for temporal alignment. The total scanning process for each mouse takes around 10 minutes.

*Second,* these 260 X-ray images are taken from variable angles are reconstructed by NRecon® with a voxel size of 7 to 10.6 μm as the **Reconstructed CT Imagery (Recon-CT)**. The 3D reconstructed CT is acquired based on the traditional filtered backprojection (FBP), which uses a 1D filter on the projection data before backprojecting (2D or 3D) the data onto the image space. Then, the 3D bone reconstructed CT is optimized to get the high quality reconstructed CT images. The reconstruction process takes over 30 minutes per mouse.

*Third,* among the Recon-CT imagery, both left and right tibiae are segmented from the knee by either our Seg-API or Dataviewer® software as the **Segmented CT Imagery (Seg-CT)**. A global threshold value (75/255) is applied and the processed images were found to agree well with gray-scaled images. The segmentation of both left and right tibiae takes around 10 to 15 minutes per mouse.

*Fourth,* after segmentation, each tibia is registered to their reference (vertically aligned tibia) by our Regist-API or Dataviewer® software as the **Registered CT Imagery (Regist-CT)**. For the first scan (week 0) of the time sequence, each segmented tibia is aligned to a reference tibia with its long axis vertical and its anterior-posterior and medial-lateral axes orthogonally positioned. Subsequent scans of the segmented tibia captured at later time points are aligned to their baseline (week 0). The registration process of each tibia to their references takes about 15 minutes.

*Fifth,* the region of interest of tibiae (proximal end where the metastatic tumors are located) overlaid to their baseline is selected by our RoI-API and the fibular are cropped manually by CTAn® as the **Region of Interest CT Imagery (RoI-CT)**. The overlapped composite RoI-CT images are generated with different pixel values assigned in each pixel based on various criterion: the pixel with bone (above threshold value of 75) both in baseline and sequential CTs are assigned to the value of 180 (light gray), the non-bone pixel (below threshold) at week 0 later became bone (above threshold) is assigned to 240 (white), the bone pixels (above threshold) at week 0 later became non-bone (below threshold) is set to 60 (dark gray), the pixel without bone in both time points were given the value to 0 (black). After the generation of CT composite, the RoI section is selected with the proximal tibia-fibula junction as the landmark, and the fibula was manually cropped by CTAn®. This process takes approximately 20 minutes for each registered tibia.

*Sixth,* the quantitative analysis of the RoI-CT imagery by CTAn®, Abaqus®, and RSA-G2 Solids Analyzer®, and the Pre-Op (Pre-Operation) and Post-Op (Post-Operation) medical records of the mice with bone metastasis are combined as the **Mice Medical Records (MiceMediRec)**. The imaging diagnosis parameters include records of tumor growth status (if available), bone structural properties, and mechanical properties. Tumor burden and growth are quantified via IVIS imaging of luminescence-tagged tumor cells, with the average radiant efficiency measured from the tibia reported. Bone structural properties, such as bone lesion occurrence, total bone volume changes, cortical bone parameters (resorption, area, and thickness), and trabecular bone parameters (bone volume fraction, number, thickness, and spacing), as well as bone mechanical properties (polar moment of inertia (pMOI)) are measured by CTAn® based on the CT in the RoI-CT dataset. For some samples, additional mechanical properties such as bone stiffness, yield strength, ultimate strength, and work to fracture are simulated or measured by finite element analysis or 3-point bending tests. This process takes over 30 minutes for each mouse.

Due to the page limit, further details on data collection and preparation processes, including breast tumor tibial inoculation and medical image processing, are provided in Section A of the supplementary materials.

## 2.2 Details of BoneMet Dataset

The total size of our BoneMet dataset is 67.87 TB, which includes six components, detailed as below. The former five components are all visual data, stored in PNG format, while the last one is numerical data, stored in CSV format.

**Rotation-X-Ray.** The Rotational X-Ray Imagery consists of 651,300 X-ray images of subjects with tumors and 676,000 X-ray images of subjects without tumors. Each image has a resolution of 4,032x4,032x1 pixels and a spatial resolution of 0.8°, captured at the hindlimb. This dataset has been aligned both spatially and temporally with the temporal resolution of 1 week, and it offers 2D X-ray images taken from multiple angles, from anterior (front) to lateral (side) and posterior (back) views, providing a comprehensive examination of the subject. The total size of this imagery is 20.93 TB. The left part of Figure 2 shows examples of the 2D X-ray images.

**Recon-CT.** The Recon-CT dataset comprises 3D CT scans reconstructed from Rotation X-Ray imagery and therefore is also temporally aligned with the temporal resolution of 1 week. The reconstruction process is illustrated in the right part of Figure 2. These slices capture cross-sectional views of the tibia, femur, and spine, with the dimensions varying according to the specific regions of interest (RoI) identified during the micro-CT reconstruction process. This component includes 2,505 CT scans of subjects with tumors and 2,600 CT scans of subjects without tumors. Each CT scan is composed of 2,685 2D slices with an image resolution of approximately 2,588x2,428x1 pixels, with an example shown in the left image of Figure 3. The total size of this dataset is 45.23 TB.

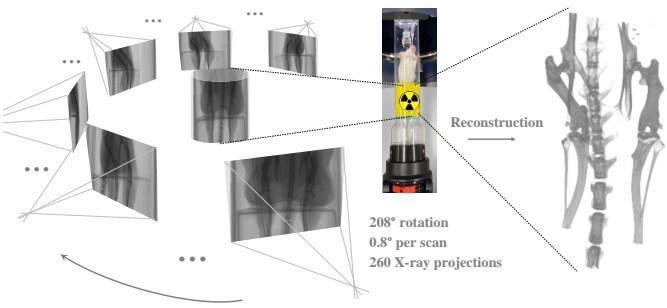

Figure 2: Examples of our Rotation-X-Ray and Recon-CT imagery. **Left:** 2D X-ray images are taken from 260 variable angles with 0.8° intervals of the hindlimb of a mouse; and **Right:** The paired 3D CT scan is reconstructed from these 2D X-ray images.

**Seg-CT.** These 3D CTs of tibiae are isolated from the 3D CT scans of hindlimb in the Recon-CT imagery, as illustrated on the right side of Figure 3. This component includes 3,005 segmented CT scans of subjects with tumors and 7,205 segmented CT scans of subjects without tumors. Each scan is composed of approximately 1,700±200 2D slices with an image resolution of approximately 700±50x900±80x1 pixels. The size of this dataset is 1.53 TB.

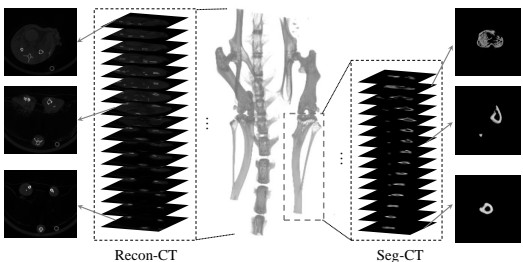

Figure 3: Examples of 3D CT scans from the Recon-CT and the Seg-CT imagery. **Left:** 3D CT scans of hindlimbs in the Recon-CT imagery; and **Right:** 3D CT scans of segmented tibiae in the Seg-CT imagery. Notably, 3D CT scans are composed of 2D cross-sectional slices.

**Regist-CT.** This imagery includes registered 3D CT scans of tibiae taken at various time points and from different animals, aligned to a reference, as shown in Figure 1 at Section A of supplementary materials. This component includes 3,005 registered CT scans of subjects with tumors and 7,205 registered CT scans of subjects without tumors. Each scan is composed of 1,538 2D slices with an image resolution of 509x539x1 pixels. The size of this dataset is 0.18 TB.

**RoI-CT.** This imagery focuses on the proximal end sections of the registered tibiae, where the effects of metastasis are most pronounced, as shown in Figure 4. The RoI-CT imagery comprises 300 2D slices below the proximal tibia-fibula junction, with overlaid registered CT scans aligned to their baseline (week 0). In each 2D slice, light gray represents the reserved bone in the sequential scans, white indicates bone formation where non-bone pixels at week 0 later became bone, and dark gray signifies bone resorption where bone pixels at week 0 later became non-bone. This component includes 3,005 CT scans of the proximal end sections of registered tibiae with tumors and 7,205 CT scans of that without tumors. Each 2D slice has the image resolution of 509x539x1 pixels. The size of this dataset is 8.00 GB.

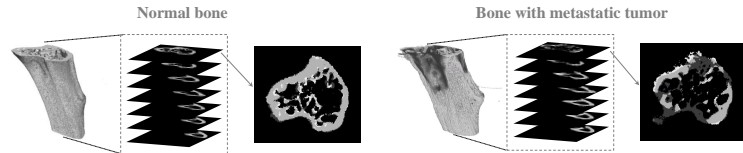

Figure 4: Examples of 3D CT scans from the RoI-CT imagery, including the proximal end of tibiae without (**Left**) and with (**Right**) metastatic breast tumors.

**MiceMediRec.** The Mice Medical Records includes detailed records for individual mice regarding animal ID, strain, date of birth, experiment data, tumor inoculation (left, right, or both tibiae), treatment type, treatment regimen (dose, frequency or duration), scan time and frequency. If available, quantitative analyses of bone from scans, FE simulations, IVIS readouts, and mechanical testing are also included to provide a comprehensive overview of the animals, their bones, and their disease conditions. More details can be found in Supplementary Materials, Section A, Table 7 . The size of this dataset is 9.44 MB.

Table 1 lists more detailed and comprehensive information for our BoneMet dataset and its six components.

Table 1: Overview of our BoneMet dataset details.

| Datasets | Size | Number of Images/Records (Positive: mouse receiving tumor injection; Negative: normal non-tumor mice. The label is assigned on the animal level, and the timing of overt bone lesion varies due to age and treatment.) | Data Format | Image Size | Anatomical Region | Temporal Resolution | Data Content |
|---|---|---|---|---|---|---|---|
| Rotation-X-Ray | 20.9 TB | Positive: 651,300 Negative: 676,000 | 2D X-ray | 4,032 x 2,688 x 1 | Hindlimb (0.8° intervals) | 1 week | X-ray images from the anterior (front view) to the lateral and posterior (back view) |
| Recon-CT | 45.2 TB | Positive: 2,505 Negative: 2,600 | 3D CT (Each with 2,685 slices) | 2,588 (±80) x 2,428 (±100) x 1 | Hindlimb | 1 week | Stacked 2D cross-sections of CT spatially and temporally paired with Rotation-X-Ray |
| Seg-CT | 1.5 TB | Positive: 3,005 Negative: 7,205 | 3D CT (Each with 1,700 (±80) slices) | 700 (±50) x 900 (±80) x 1 | Tibiae | 1 week | The segmented tibiae from Recon-CT |
| Regist-CT | 0.2 TB | Positive: 3,005 Negative: 7,205 | 3D CT (Each with 1,538 slices) | 509 x 539 x 1 | Tibiae | 1 week | The registered tibiae dataset aligned to a vertically aligned reference |
| RoI-CT | 8.0 GB | Positive: 3,005 Negative: 7,205 | 3D CT (Each with 300 slices) | 509 x 539 x 1 | Proximal end of tibiae | 1 week | The selected CT from region of interest (RoI) of bone metastasis |
| MiceMediRec | 9.4 MB | Positive: 501 Negative: 520 | Text | NA | NA | 1 week | Medical record and quantitative analysis results of 3D CT images, simulation and mechanical testing |

## 2.3 POTENTIAL APPLICATIONS

There are many applications can be supported by BoneMet database. Due to page limits, we only validate some critical applications in Section 3, including i) 2D spatial-temporal X-ray imagery-based BCBM diagnosis; ii) 3D CT imagery-based BCBM diagnosis; and iii) 3D multi-modal RoI-CT imagery-based BCBM prognosis; and iv) Sparse-angle 3D CT reconstruction with one real 2D X-ray images. Besides, more supportive applications, such as 3D CT auto-segmentation, generalist biomedical AI diagnosis, among others, are listed in Supplementary Materials, Section A, Table 6 to exhibit its broader applicability.

## 3 EXPERIMENTS AND RESULTS

We conduct experiments on our BoneMet dataset via developing various deep learning solutions to exhibit its applicability and efficiency to manage BCBM disease. Currently, the positive and negative labels are assigned at the animal level, rather than individual time points and individual X-ray images. A positive label of a mouse indicates that a metastatic bone lesion occurs in the subject between week 0 and week 5. There are animal-to-animal variations in the times of bone lesion initiated and the speed of lesion growth.

### 3.1 HYPERPARAMETERS AND DATASET SPLITTING

The hyperparameters used in our Breast cancer bone metastasis (BCBM) diagnosis experiments follow the common practice of supervised ViT training. The key hyperparameters are detailed in Table 2.

For the experiments presented in our manuscript, we employ a train/test split with a ratio of 8:2. Specifically, 80% of the mice were used for training the model, and the remaining 20% were reserved for testing. Within the training dataset, we implemented a 5-fold cross-validation strategy.

This process is repeated five times, with each subset serving as the validation set once, so the validation data is never seen by the model during the training phases, ensuring that there is no data leakage. In our trials, the data split is conducted at the mice level, so the entire sets of images from individual mice were allocated either to the training or testing datasets. This is to ensure no overlap of data from the same mice between the training and testing datasets, preventing any potential data leakage. All our experiments are conducted on a high-performance workstation equipped with an NVIDIA RTX A6000 GPU, which has 48GB of VRAM.

Table 2: Training Configuration

| Config | Value |
| --- | --- |
| Optimizer | AdamW |
| Base learning rate | 1e-3 |
| Weight decay | 0.05 |
| Optimizer momentum | $\beta_1, \beta_2 = 0.9, 0.999$ |
| Learning rate schedule | Cosine decay |
| Warmup epochs | 5 |
| Training epochs | 100 |
| Augmentation | RandAug (9, 0.5) |

## 3.2 APPLICABILITY OF OUR ROTATION-X-RAY DATASET FOR BCBM DIAGNOSIS

We conduct experiments to demonstrate the applicability of our Rotation-X-Ray dataset to manage BCBM disease. Here, we employ two Vision Transformers (ViT) variants for the diagnosis of BCBM using Rotation-X-Ray imagery: a simple ViT variant Swin-Base (25), and a spatial-temporal ViT variant MMST-ViT (21). Notably, the 2D X-ray images within the Rotation-X-Ray imagery have been carefully spatially and temporally aligned. For clarity, we denote MMST-ViT as ViT (w/ STA), indicating its ability to leverage spatial-temporal alignment (STA), and Swin-Base, as ViT (w/o STA), indicating not utilizing STA. Table 3 presents the performance outcomes measured by the metrics of Precision, Recall, F1-Score, and Accuracy. We have two observations. First, both ViT (w/o STA) and ViT (w/ STA) can achieve decent diagnosis performance results, with the overall accuracies of 79.1% and 89.0%, respectively. This suggests that our dataset is adaptable to various ViT variants, from simple to more complex architectures, for effective BCBM diagnosis. Second, we observe a significant training-test accuracy gap of 20.4% with ViT (w/o STA), indicating a pronounced overfitting issue inherent to the ViT architecture. In contrast, the incorporation of STA in the Rotation-X-Ray imagery substantially alleviates this issue, as demonstrated by the notably smaller training-test accuracy gap of 6.0% achieved by ViT (w/ STA). This highlights the effectiveness of STA in mitigating overfitting and enhancing model generalizability on our Rotation-X-Ray dataset.

Table 3: The BCBM diagnosis using ViT with and without STA on the Rotation-X-Ray imagery

| Methods | Training | | | | Test | | | |
| --- | --- | --- | --- | --- | --- | --- | --- | --- |
| | Precision | Recall | F1-Score | Accuracy | Precision | Recall | F1-Score | Accuracy |
| ViT (w/o STA) | 99.6 | 99.7 | 99.7 | 99.5 | 92.3 | 80.5 | 86.0 | 79.1 |
| ViT (w/ STA) | 95.4 | 96.9 | 96.1 | 95.0 | 92.1 | 90.6 | 91.3 | 89.0 |

## 3.3 APPLICABILITY OF 3D CT SCANS FOR BCBM DIAGNOSIS ACROSS VARIOUS MODEL ARCHITECTURES

Here, we conduct experiments by utilizing 3D CT scans in the Regist-CT imagery for BCBM diagnosis. We utilize two CNN-based model architectures—BigTransfer (BiT-M) (16) and EfficientNetV2-M (35)—along with one ViT-based model architecture, *i.e.*, Swin-B, as the backbone networks. These models extract slice-level features from each 2D slice within the CT volume and subsequently aggregate these features into volume-level representations using a max-pooling layer.

Figures 5a, 5b, 5c, and 5d present the performance results. We observe BiT, EfficientNetV2, and Swin-B exhibit commendable performance, achieving overall accuracies of 81.0%, 85.0%, and 92.0%, respectively. These results validate that our BoneMet dataset is compatible with both CNN-based and ViT-based model architectures, demonstrating its broad applicability. Furthermore, the Swin model consistently outperforms its two CNN-based counterparts. For instance, in terms of the F1-Score, it exceeds BiT and EfficientNetV2 by 9.3% and 5.9%, respectively. This superior performance is attributed to the ViT-based model's use of Multi-head Self-Attention (MSA) (38), which enhances its ability to effectively aggregate slice-level features into volume-level representations.

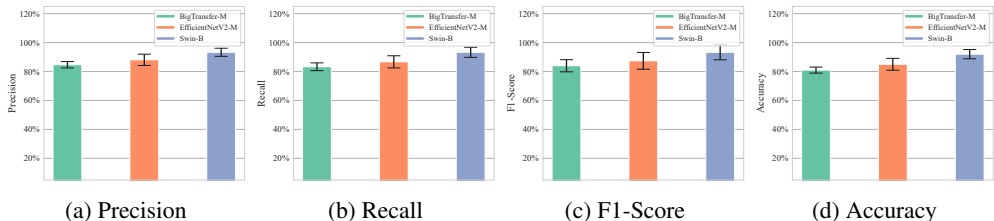

(a) Precision  (b) Recall  (c) F1-Score  (d) Accuracy

Figure 5: The diagnosis of BCBM using 3D CT scans from the Regist-CT imagery.

## 3.4 APPLICABILITY OF RoI-CT IMAGERY AND MICEMEDIREC DATASETS FOR MULTI-MODAL PROGNOSTIC ASSESSMENT OF BONE MECHANICAL PROPERTIES

Managing pathological fractures associated with bone metastases is crucial for preserving a patient's mobility and quality of life. In this context, we explore the applicability of our RoI-CT dataset to this critical scenario by assessing the mechanical competence of the bone at a future time point. We formulate this task as a multi-modal prognostic assessment of bone mechanical properties. First, the 3D CT scans in the RoI-CT imagery are used to train generative models to produce 3D CT scans of future frames, reflecting the progression of bone lesions. Next, the biological data in the MiceMediRec component is utilized to simulate the mechanical behaviors of axial compression of the proximal end of tibiae with metastatic osteolysis.

Three generative models are taken into account: 3D Generative Adversarial Networks (3D-GAN) (2), Temporal Variational Autoencoders (T-VAE) (43), and Spatial-Temporal Variational Autoencoders (ST-VAE), for generating future 3D CT scans. The quality of these generated scans is measured using PSNR (Peak Signal-to-Noise Ratio), SSIM (Structural Similarity), and LPIPS (Learned Perceptual Image Patch Similarity) metrics. Note that a higher PSNR (and SSIM) value or a lower LPIPS value indicates better generation quality. Table 4 presents the quantitative results. We observe that all three methods achieve high-quality future CT generations. For instance, the 3D-GAN, T-VAE, and ST-VAE methods achieve SSIM values of 0.767, 0.817, and 0.860, respectively.

Table 4: Evaluations of the quality of 3D CT prediction using three metrics

| Methods | PSNR ($\uparrow$) | SSIM ($\uparrow$) | LPIPS ($\downarrow$) |
|---|---|---|---|
| 3D-GAN | 21.9 | 0.767 | 0.098 |
| T-VAE | 23.4 | 0.817 | 0.078 |
| ST-VAE | 35.5 | 0.860 | 0.041 |

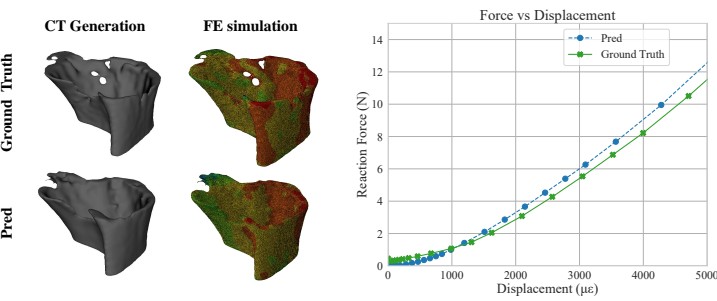

(a) CT generation and FE simulation  (b) Reaction force prediction

Figure 6: Evaluations of prognostic assessment of bone mechanical properties: **(a)** comparisons between ST-VAE-generated 3D CT scan and its ground truth, along with the corresponding finite element (FE) analysis of axial compression of proximal tibiae; and **(b)** comparisons between the predicted and ground truth reaction force values at various displacement levels.

The visualization and quantitative results obtained using the ST-VAE model shows the quality of 3D CT prognosis, as exhibited in Figure 6. In Figure 6a, the predicted 3D rendering volume is comparable to the ground truth, and the finite element (FE) simulations of axial compression on meshes constructed from the predicted bone structures closely resemble those constructed from the ground truth. Furthermore, Figure 6b illustrates the curves of the ground truth and predicted reaction force values at various displacement levels. It is evident that the predicted reaction force curve closely

matches the ground truth, achieving an impressive $R^2$ value of 0.956. These results underscore the significance of our BoneMet data in the multi-modal prognostic assessment of bone mechanical properties with metastatic bone lesions.

### 3.5 APPLICABILITY OF SPARSE-ANGLE CT RECONSTRUCTION

Conventional CT reconstruction methods require 2D X-ray images from multiple angles. For example, using NRecon® software necessitates at least 260 X-ray images from different angles for accurate reconstruction. This leads to prolonged exposure to ionizing radiation during the process of capturing multiple 2D X-ray images, incurring severe adverse effects on the patient's health. Here, we explore the applicability of our Rotation-X-Ray dataset to sparse-angle CT reconstruction, which can be validated by paired CTs in Recon-CT dataset. This task can significantly reduce radiation exposure by reconstructing 3D CT scans with continuous viewpoint rotations from a single 2D X-ray image.

We have employed two NeRF-based methods: PixelNeRF (45) and MedNeRF (4), to reconstruct 3D CT scans. The reconstruction quality is measured by the PSNR (Peak Signal-to-Noise Ratio), SSIM (Structural Similarity Index), FID (Fréchet Inception Distance), and KID (Kernel Inception Distance) metrics. Notably, higher PSNR and SSIM values, along with lower FID and KID values, indicate better reconstruction quality. Table 5 presents the quantitative results of reconstructing 3D CT scans. It is observed that our dataset can support both NeRF-based methods for sparse-angle CT reconstruction. For example, the MedNeRF can achieve superb reconstruction quality, with a high PSNR (and SSIM) value of 30.2 (and 0.810) and a low FID (and KID) value of 91.5 (and 0.092). Moreover, we use the MedNeRF method to generate a complete set of CT projections within a full vertical rotation from a given single-view X-ray of a CT slice, with results shown in Figure 7. Despite the challenge of this task, the MedNeRF method produces high-quality reconstructions, demonstrating the applicability of our BoneMet dataset for sparse-angle CT reconstruction.

Table 5: Quantitative evaluations of sparse-angle CT reconstruction

| Methods | PSNR (↑) | SSIM (↑) | FID (↓) | KID (↓) |
|---------|----------|----------|---------|---------|
| PixelNeRF | 20.2 | 0.740 | 155.2 | 0.128 |
| MedNeRF | 30.2 | 0.810 | 91.5 | 0.092 |

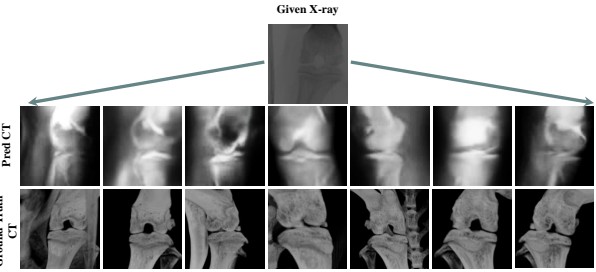

Figure 7: Illustration of sparse-angle 3D CT reconstruction from a 2D X-ray image by MedNeRF method. The first row also presents the given 2D X-ray image and X-ray images at other angles for reference. The second row shows the reconstructed 3D CT scans, while the third row displays the ground-truth 3D CT scans from the Recon-CT imagery.

We have also conducted additional experiments, including tibiae auto-segmentation, to validate the significance and broad applicability of our Recon-CT and Seg-CT dataset, and generalist biomedical AI diagnosis of metastatic breast tumor on bone, to demonstrate the applicability of our RoI-CT and MiceMediRec dataset. The details of these experiments are deferred to Sections B of the supplementary materials. Besides, the future potential applications of the BoneMet dataset are discussed in Sections D of the supplementary materials.

## 4 THE BONEMET PACKAGE

In addition to our BoneMet dataset, we also develop the BoneMet package, including three types of APIs: 1) CT Image Segmentation, 2) CT Image Registration, and 3) RoI-based CT Image Cropping), at the Python Package Index(PyPI), for public release to facilitate our dataset's ease access. The details of three APIs and their usage examples are deferred to Section C of supplementary materials.

## 5 CONCLUSIONS

This paper introduces the BoneMet dataset, the first large-scale meticulously curated collection of well-organized CT and X-ray images in angles, positions and time points designed specifically for the diagnosis and prognosis of BCBM disease. It includes high-resolution images that are well-organized across spatial and temporal dimensions, allowing for an in-depth analysis of bone responses to tumor progression and treatment. The integration of detailed imaging with medical metadata enhances the utility of the BoneMet dataset, enabling the development and validation of advanced deep learning models. This dataset is also invaluable for developing high-quality CT reconstruction techniques from sparse X-ray data and for performing 3D segmentation. Our extensive experimental evaluations confirm that the dataset is compatible with various deep learning approaches. Alongside the dataset, we have developed the BoneMet Package to assist researchers and practitioners in automatic Breast cancer bone metastasis medical images processing such as 3D CT images segmentation, registration and region of interest (RoI) selection, as well as constructing their own deep learning models. While the primary aim of the BoneMet dataset is to advance deep learning models in disease diagnosis, management and medical image processing techniques, its potential applications extend across medical imaging, oncology, and computational pathology. We believe that our BoneMet dataset will be a significant valuable asset to the fields of deep learning, medical radiology, orthopedics and oncology, spurring further interdisciplinary research at the intersection field of healthcare and artificial intelligence.

## 6 BROADER IMPACTS

The ultimate goal of our project is to unleash the power of deep learning in BCBM diagnosis and prognosis. In this study, we have attempted to attack the major obstacle—the scarcity of high-resolution image datasets of early-stage BCBM with sufficient diagnostic and prognostic labeling—with our open-sourced large-scale BoneMet datasets based on murine models. Utilizing our datasets, we have demonstrated the feasibility of generating multi-angle x-ray images of tibiae and 3D CT reconstructions from sparse-angle x-rays. These generative 3D reconstruction techniques developed using our murine dataset could be refined and applied to human patient x-ray images which are more commonly used for screening. We believe that this regenerative approach will address the radiation exposure concerns and help develop human breast cancer bone metastasis datasets that cover all disease stages, especially the early phases and include sequential and safe imaging of the patients. We anticipate that the human datasets would open the door to build and validate foundation models for human BCBM. The two tasks could potentially form a positive feedback loop: a high-quality dataset drives the performance of AI learning of more accurate and predictive foundation models, which in turn increases the efficacy of the generative 3D reconstruction algorithm, leading to further successful expansion of the dataset.

We wish to engage AI scientists, biomedical investigators, clinicians, and policymakers in developing more effective diagnosis, prognosis, and treatments of cancer bone metastasis. By publishing our BoneMet dataset, we envision that more stakeholders can benefit from full access to the dataset, which could be used to develop early diagnostic tools for breast cancer metastasis with improved accuracy and sensitivity. The dataset and the insights it generates could play a crucial role in advancing AI applications, such as large vision models and multimodal approaches with temporal and spatial alignment, for the early diagnosis and prognosis of breast cancer bone metastasis, particularly in enhancing the predictive accuracy of models used in clinical settings when the disease progression is relatively slower. As we continue to refine and expand our dataset, we aim to provide a robust resource for better understanding, detection, and treatment of bone metastases in human breast cancer, and further support future innovations in deep learning studies and clinical practices.

## 7 ACKNOWLEDGMENTS

This work was partially funded by the following grants: Delaware Center for Musculoskeletal Research (DCMR, P20GM139760) Research Core Access Award (to L. Wang), Women's Health supplement (P20GM139760 - 03W1, to L. Wang), and DCMR pilot project (to X. Yuan). Any opinions and findings expressed in the paper are those of the authors and do not necessarily reflect the view of funding agencies.

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

## OUTLINE

This document provided supplementary materials to support our main paper. Section A provides details of Breast cancer bone metastasis model creations and data collection processes. Section B presents additional experimental settings and results. Section C lists three examples of API for CT Image Segmentation, Registration and RoI-based CT Image Cropping, and section D talks about future potential application of our BoneMet dataset.

## A    DETAILS OF DATASET AND DATA COLLECTION

In order to build such large-scale BoneMet dataset, the team has spent more than five years with three to four graduate students assisted with over ten undergraduates and summer students. The experimental expenses include mice purchase, shipping, per diem, surgical tools, cell culture, and user fee to access micro-CT scanners, mechanical testers, microscopy. Usually, it takes approximately one year's training to become an expert in inoculating cancer to mice and 2 to 3 month each batch of the mice (around 20 mice) for data acquisition, image processing including CT reconstruction, segmentation, registration, and image analysis.

**Image Acquisition**: For each scan, mice were anesthetized with 3% (v/v) isoflurane and held in the built-in holder, which was rotated 208-degree with a step of 0.8 degrees and there are 260 X-ray images taken in total. One frame was taken per step with the following settings: 900 ms exposure time, X-ray of 200 mA current and 50 kVp, and a 0.5 mm Al filter, cone-beam angle Horizontal of 25.888290 (deg) and Vertical 17.423092 (deg). The in vivo scanning of the hindlimbs lasted around 4 min per animal and the accumulated radiation exposure (< 600 mGy) was low.

**3D Reconstruction**: The 3D volume or 2D cross-sectional images of micro-CT were reconstructed from 260 X-ray projections using the NRecon® software (Bruker) with a voxel size of 7 to 10.6 μm. The 3D reconstructed CT is acquired based on the traditional filtered backprojection (FBP),which uses a 1D filter on the projection data before backprojecting (2D or 3D) the data onto the image space. Then, the 3D bone reconstructed CT were optimized with the several steps such as center of rotation adjustment,beam hardening correction,ring artifact reduction and reconstruction filtering in order to get the reconstructed micro-CT images with high quality. Among these steps, center of rotation adjustment is used to ensure that the reconstruction is symmetrical, beam hardening correction is to compensate artifacts in the scan. Since the 260 X-ray images were taken by the rotation of cameras, the ring artifacts generated during the reconstruction process is minimized by ring artifact reduction. The median filter is employed to remove the salt-and-pepper noise in the reconstructed images, which is the common noise happening in the medical images.

**Segmentation**:The the left and right tibiae were segmented from reconstructed CT images by our python segmentation package. A global threshold value (75/255) was applied and the processed images were found to agree well with the gray-scaled images. Two limbs of each mice were separated by their momentum intensity in their relative positions and the tibia and femur of each limb were segmented by identify the specific structure of the knee, where the position of the knee is found by comparing the cross-sectional areas of each 2D slices of the bone and the minimal is where the knee located.

**Registration (alignment)**: After the segmentation, the week 0 scans of each mice tibia were aligned to a registration reference tibia CT, where the long axis positioned vertically and the anterior-posterior and medial-lateral axes arranged orthogonally, and the transformation and rotation were manually adjust by Dataviwer®. Then the registered tibiae CT at week 0 scans served as baseline scan for

their subsequent week scans. The mutual information maximizing between those of tibiae CT from different mice at week 0 and the registration reference tibia CT, and between those of the sequential scans and themselves week 0 baselines. The mutual information is a measure of the statistical dependence or information shared between the image intensities of the two images. Before transformation (rotation, translation, scaling, *etc.*), initial coarse alignment is performed to reduces the search space for the registration algorithm, then transformation applied to the moving image, maximizing the mutual information between the fixed image and the transformed moving image by gradient descent optimization. Once the optimal parameters are found, the final transformation will be applied to the moving image to align it with the fixed image.

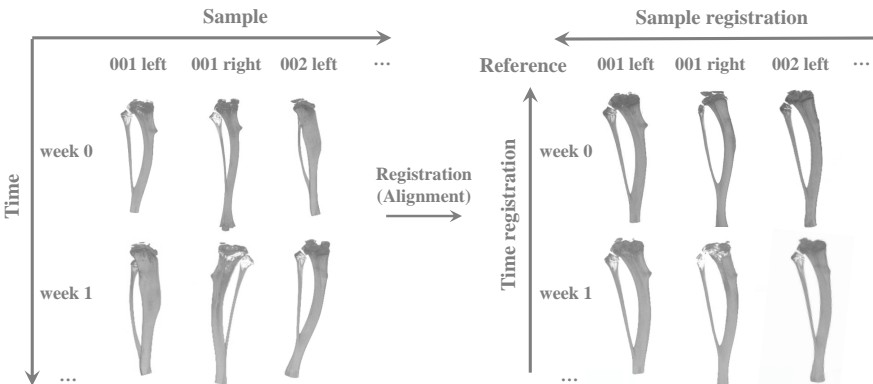

Figure 8: Examples of 3D CT scans from the Seg-CT and the Regist-CT imagery. **Left:** 3D CT scans from the Seg-CT imagery (*i.e*, w/o registration); and **Right:** 3D CT scans from the Regist-CT imagery, which are registered relative to their vertically aligned reference at various time points and across different animals.

**Selection of Region of Interest**: The overlapped composite RoI-CT images were generated with different pixel values assigned in each pixel based on various criterion: the pixel with bone (above threshold value of 75) in baseline and sequential scans were assigned to the value of 180 (light gray), the non-bone pixel (below threshold) at week 0 later became bone (above threshold) is assigned to 240 (white),the bone pixels (above threshold) at week 0 later became non-bone (below threshold) is set the value to 60 (dark gray), the pixel without bone in both time points were gave the value to 0 (black). After the generation of CT composites, the RoI section was selected automatically with the proximal tibia-fibula junction as the landmark, where the number of contoured circles of bone tissues in each 2D slice changes. Then, the fibula in the 2D slices was manually cropped by CTAn® with interpolated mask.

**Analysis and Quantification**: The desired analysis such as bone volume changes was conducted on the overlaid tibiae ROI or registered CT to access the bone structure morphology changes, metastatic breast cancer disease progression and mechanical properties impairment over time with and without treatment effects. Cortical polar moment inertia (Ct.pMOI), bone mineral density (Ct.BMD), and tissue mineral density (Ct.TMD) as well as trabecular bone volume fraction (Tb.BV/TV), thickness (Tb.Th), separation (Tb.Sp), bone mineral density (Tb.BMD), and tissue mineral density (Tb.TMD) were calculated using CTAn® 3D analysis software. The overlaid subsequent scans with the baseline scan (Week 0) of the same tibia make it possible to quantify the changes (Delta values) of each tibia at subsequent weeks relative to week 0.

## B    SUPPORTING EXPERIMENTAL SETTINGS AND RESULTS

### B.1    TIBIAE AUTO-SEGMENTATION

Conventional nnU-net as well as emerging foundation models such as MedSAM have demonstrated the improved accuracy and robustness for universal medical image segmentation (26). TotalSegmentator, which is pretrained on the nnU-Net segmentation algorithm with the dataset of 1204 CT examinations, presents a deep learning model to automatically and robustly segment all major

Table 6: Overview of the six components in our BoneMet dataset

| Components | Advantages | Applications |
|---|---|---|
| **Rotation-X-Ray** | Low cost, wide availability, and minimal radiation exposure | Pre-training and fine-tuning deep learning models for bone metastasis early detection and monitoring |
| **Recon-CT** | Reveal the precise location and size of bone lesions with detailed 3D structure and 2D cross-sections | Training deep learning models for bone metastasis localization, segmentation, and sparse angle reconstruction |
| **Seg-CT** | Enhance the diagnosis efficiency by segmenting limited regions | Training deep learning models for bone segmentation and region-specific analysis of metastatic lesions |
| **Regist-CT** | Increase sensitivity, making subtle changes in the bone structures detectable | Training deep learning models for longitudinal analysis, early diagnosis, prognosis of metastatic changes, disease progression and treatment effects analysis |
| **RoI-CT** | Enable the precise diagnosis and quantitative studies | Training deep learning models for quantitative analysis and prediction of metastatic bone structural and mechanical properties |
| **MiceMediRec** | Detailed demographic information about the animals and the disease | Enabling multimodal deep learning models for comprehensive disease diagnosis, prognosis, and generalist biomedical AI diagnosis |

Table 7: Details of MiceMediRec Dataset

| Source | Parameters | Description |
|---|---|---|
| Medical record | Age | Unit: week |
| | Sex | |
| | Date | The date of cancer inoculation |
| | Body Weight | Unit: g |
| | Tibiae cancer inoculation | Records of each tibiae with cancer or without cancer |
| | Treatment | Treatment to metastatic breast cancer, like chemotherapy. Dose Unit: $\mu L/g$ |
| | Diagnosis | Diagnosis to bone metastasis, like bone lesion |
| Quantitative analysis | Average Radiant Efficiency | IVIS signal to breast cancer in the bone. Unit: $\mu W/cm^2$ |
| | Bone volume | Bone total volumes. Unit: $mm^3$ |
| | Cort. pMOI | Cortical bone polar moment of inertia. Unit: $mm^4$ |
| | Cort. Ar | Cortical bone area. Unit: $mm^2$ |
| | Cort. Th | Cortical bone thickness. Unit: mm |
| | Trab. BV/TV | Trabecular bone volume relative to the marrow volume. Unit: % |
| | Trab. Th | The average thickness of the individual trabeculae. Unit: mm |
| | Trab. N | The number of trabeculae per unit length. Unit: $mm^{-1}$ |
| | Trab. Sp | The average distance between trabeculae. Unit: mm |
| | Displacement | the deformation of the bone in response to applied load. Unit: $\mu m$ |
| | Reaction force | The force exerted by a constraint on the bone in response to an applied load. Unit: N |
| | Stiffness | Bone resists deformation in response to an applied force. Unit: N/m |
| | Yield load | The force where the bone begins to deform permanently. Unit: N |
| | Maximum load | The highest force the bone can safely withstand before failure. Unit: N |
| | Work to fracture | The energy bone absorbed before fracture. Unit: N·m |

anatomic structures on body CT images (41). In this experiment, we use the TotalSegmentator to segment the right tibia from hindlimb in our Recon-CT dataset with variable resolutions by downsizing, and employ our paired Seg-CT dataset as the ground truth. As shown in Figure 9, the left side is the depiction of the 3D render of hindlimb CT from the Recon-CT dataset. After inference, the nnU-Net model successfully identified the left tibia from other part of the hindlimb, including the femur, spine, and hips in all CTs with variable resolutions and most of the tibiae are segmented accurately by the TotalSegmentator, with better segmentation in higher resolution CT when compared with the ground truth tibia from our Seg-CT dataset. This experiment underscores the importance of high resolution dataset for effective bone segmentation. The segmentation of the tibia using the nnU-Net deep learning model hold significant potential for applications in disease characterization, surgical and radiation therapy planning.

## B.2 GENERATIVE AI MODEL FOR DIAGNOSIS OF METASTATIC BREAST CANCER ON BONE

Generative AI (GenAI) models can flexibly interpret different combinations of medical modalities, including data from imaging, electronic health records and laboratory results, and in turn produce expressive outputs for disease diagnosis assistance with advanced medical reasoning abilities (28). In this experiment, we used a LLaVA (Large Language and Vision Assistant), which is a combination of the cutting-edge LLaMa 2 text generator and OpenAI's CLIP for image embedding, and fine-tuned

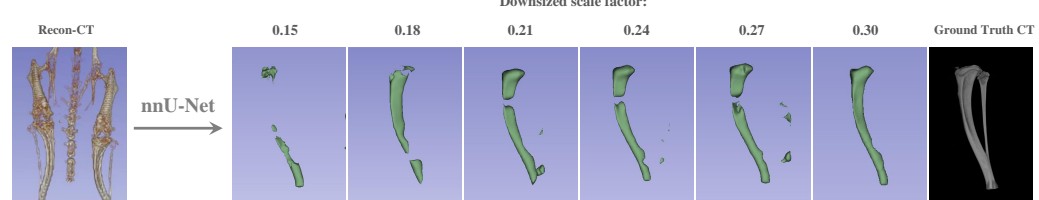

Figure 9: Illustration of tibiae 3D CT auto-segmentation on the Recon-CT dataset from low (left) to high (right) resolutions by adjusting the downsizing scale factor. The right tibia is segmented by nnU-Net. Compared with the ground truth, the TotalSegmentator performs better on high resolution CTs.

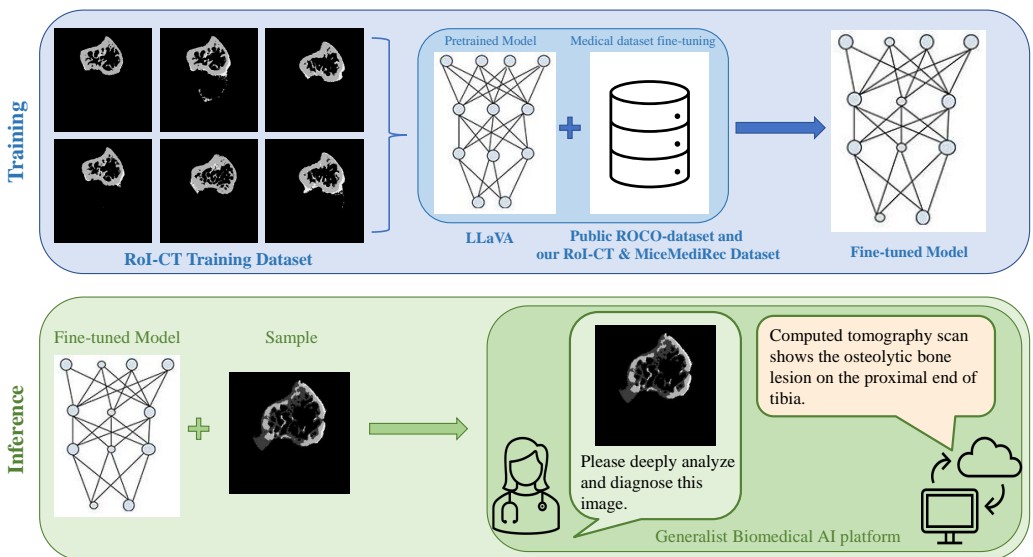

Figure 10: Illustration of the training and inference process, and Generative AI platform for diagnosis of breast cancer bone metastasis given a 2D cross-section image from RoI-CT dataset.

with ROCO-dataset (the medical text-image pairs) and our RoI-CT imagery with MiceMediRec text dataset pairs, to generate CT analysis to assist Breast cancer bone metastasis diagnosis (19) (27). As shown in Figure 10, the Web interface demonstrates the ability of fine-tuned LLaVA model to generate the diagnosis given a 2D cross-sectional image with breast cancer bone metastasis from RoI-CT dataset.

We have justified the selection of benchmark methods based on their relevance and performance in similar tasks before, as well as their compatibility with the BoneMet dataset. Additionally, we plan to include more comparative analyses with alternative methods in future work to further evaluate the dataset and benchmarks.

# C  THE DETAILS OF THREE APIS AND THEIR USAGE EXAMPLES

## C.1  CT IMAGE SEGMENTATION.

This API provides a simple interface to segment the 3D Reconstructed CT (Recon-CT) images into separate CT scans for the spine, left tibia, left femur, right tibia, and right femur. It can handle individual or batched segmentation of the Recon-CT scans. The API reads the 3D CT scans, identifies the appropriate indices to split the images, and saves the segmented scans to the specified output paths. Given the time point (e.g., the week after cancer inoculation), the input folder path, and the

output folder path, Figure 11 exhibits how to utilize the CT Image Segmentation API to automatically segment the tibiae.

```
config = {
    "week": " week 0",
    "masterfolder": r"F:\Recon-CT\week 0",
    "masterout": r"F:\Seg-CT\week 0"
}

splitter = ReconCTSegmentation(config)

# Split a single image
input_folder = r"F:\Recon-CT\week 0\871"
image_title = "871"
splitter.split_image(input_folder, image_title, config["masterout"])

# Split multiple images
for folder in os.listdir(config["masterfolder"]):
    if folder[0:10] in [871, 872, 873, ...]:
        input_folder = os.path.join(config["masterfolder"], folder)
        image_title = os.path.basename(folder)[0:12]
        splitter.split_image(input_folder, image_title, config["masterout
    "])
```

Figure 11: Example of our CT Image Segmentation API.

```
config = {
    "workspace": r"F:\Seg-CT\week 0",
    "outputdir": r"F:\Regist-CT\week 0",
    "refdir": r"F:\reference",
    "img_z_range": [None, None],
    "ref_z_range": [None, None],
    "initial_transform_angles": [np.pi * i / 16 for i in range(-16, 10)],
    "BASELINE_REG": True, # week 0 (True) or sequencial scans (False)
}

# Initialize the registration instance
registration = CTRegistration(config)

# Register a single CT scan
input_folder = r"F:\Seg-CT\week 0"
ct_id = "871 week 0 left tibia"
week = 0
output_folder = config["outputdir"]
registration.register_ct(input_folder, ct_id, week, output_folder)

# Register a batch of CT scans
input_folder = r"F:\Seg-CT\week 0"
ct_ids = ["871 week 0 left tibia", "871 week 0 right tibia", "872 week 11
    left tibia", ...]
week = 0
output_folder = config["outputdir"]
registration.batch_register(input_folder, ct_ids, week, output_folder)
```

Figure 12: Example of our CT Image Registration API.

## C.2 CT IMAGE REGISTRATION.

This API helps researchers with the tibia registration on Seg-CT dataset. It can handle individual or batched registration of the segmented tibiae CTs. The API loads the reference and target CT scans, performs initial transformation, and registers the target CT scan to the reference CT scan. Then the registered CT scan and the transformation are saved to the specific output folder. Given the time point

```python
# Configuration
config = {
    "foldername": "selected 300 slices below proximal Tibia-fibular
    junction",
    "first_slice_selected": "first slice selected",
    "last_slice_selected": "last slice selected",
    "first_slice_selected_below_t-f_junction": 0  # Index of the first
    selected slice below the tibia-fibular junction
}

# Initialize the RoICropper
cropper = RoICompositeCropper(config)

# Crop the RoI from CT images
input_folder = r"F:\Regist-CT\Tibia w0w5composite"
output_folder = os.path.join(input_folder, config["foldername"])
first_slice_selected = config["first_slice_selected"]
last_slice_selected = config["last_slice_selected"]
first_slice_below_tf_junction = config["first_slice_selected_above_t-
    f_junction"]

cropper.crop_roi(input_folder, output_folder, first_slice_selected,
    last_slice_selected, first_slice_below_tf_junction)
```

Figure 13: Example of our RoI-based CT Image Cropping API.

(e.g., the week after cancer inoculation), the slices range of reference and target subjects, the input folder path, the reference folder path, and the output folder path, Figure 12 illustrates how to utilize the CT Image Registration API to automatically align the segmented tibiae.

### C.3 RoI-based CT Image Cropping.

This API provides a simple interface to crop the region of interest (tibia proximal end) on Regist-CT dataset. It can handle batched cropping of the Regist-CT dataset. The API reads the overlapped 3D Regist-CT composite processed by our python package, identifies the proximal tibia-fibular junction, selects appropriate indices to split the images, and saves the cropped to the specified output paths. Given the input folder path, the output folder path, and index of the first selected slice below the tibia-femoral junction, Figure 13 demonstrates how to utilize the RoI-based CT Image Cropping API to automatically crop the proximal end of tibiae.

## D  BAD-CASE ANALYSIS

Regarding to the bad case analysis, we conducted the following analyses: label noise and data imbalance. Label noise is not a concern in our primary dataset, as positive and negative labels are assigned at the animal level. However, we manually created a separate dataset with noisy labels. To address label noise in this separate dataset, we applied Label Smoothing (34) with a smoothing factor of 0.1, following the DeiT approach (37). The model's accuracy improved from 71.43% (without Label Smoothing) to 95.86% (with Label Smoothing), demonstrating the technique's effectiveness in reducing errors associated with noisy labels.

We also examined the effects of data imbalance in our dataset, which had a negative-to-positive ratio of 5:1. To mitigate the challenges posed by this imbalance, we implemented Focal Loss (22). The model trained with Cross-Entropy Loss achieved an accuracy of 66.67%, whereas the model trained with Focal Loss improved the accuracy to 69.36%.

## E  FUTURE POTENTIAL APPLICATION OF BONEMET DATASET

The BoneMet dataset also holds other significant promise for advancing various applications. For example, it can be leveraged to develop foundation models and self-supervised contrastive learning

techniques (49), which will enhance the model's ability to learn robust and generalizable features from the dataset without extensive labeled data. Additionally, the dataset can facilitate the prediction of multi-angle X-ray images, providing a comprehensive view of bone metastases from different perspectives with reduced radiation exposure and improving diagnostic accuracy and aid in better visualization of complex anatomical structures. Moreover, the BoneMet dataset can be utilized for finite element analysis (FEA) prediction of metastatic bone mechanical properties (31). By integrating FEA with deep learning models, researchers and clinic doctors can easily predict how metastatic lesions affect the mechanical integrity of bones without running the complicated and time-consuming finite element simulation, which is crucial for the broad application of finite element analysis in assessing fracture risk and planning appropriate treatments. Lastly, the dataset can be used to train 3D CT registration models, such as the Convolutional Neural Networks (ConvNets) and Deep Learning Image Registration (DLIR) framework (7). These models can accurately align 3D CT scans over time or across different imaging modalities, enabling precise monitoring of disease progression and the effectiveness of treatments.

## F ETHICAL STATEMENT

All the animal procedures including cancer implantation and micro CT scans have been approved by the authors' Institutional Animal Care and Use Committee (IACUC). We made every effort to minimize animal suffering throughout the research process. In brief, the pain and stress associated with cancer implantation and cancer growth were carefully monitored (via signs of inflammation, loss of body weight, changes of normal behaviors, and cancer burden) and managed (via administration of painkillers and humane sacrifice). Mice were anesthetized using 3% isoflurane gas, ensuring a deep, stable sleep state that prevented movement during surgery and the imaging process with minimized stress or discomfort. Additionally, to reduce radiation exposure, each mouse received 5 weekly micro CT scans of lower limbs using a Bruker/Skyscan in vivo scanner (Bruker 1276). The advantage of this scanner vs. other commercially available in vivo scanners is the low radiation exposure during each session (< 500 mGy) due to the faster scan speed (900 ms) and fewer projections (260). The radiation dose (< 0.5 Gy) is much lower than the lethal radiation dose for mice (10.5 Gy). Our BoneMet dataset adheres to all relevant regulations, including those concerning animal welfare and data protection. Furthermore, we are committed to following guidelines for ethical AI development, including ensuring transparency in AI model development, avoiding bias, and maintaining data privacy, ensuring that the use of this dataset aligns with best practices in responsible AI research.

## G LIMITATIONS

Although animal models of breast cancers provide valuable insights into disease development and allow testing the accuracy and sensitivity of various diagnosis methods, we acknowledge the limitations of using animal models in general and the specific model adopted in our study. One potential limitation is the difference in physiology and immune response between mice and humans, leading to the dramatic divergence of the disease progression speed (weeks in mice vs. years in humans). The other limitation is that the mice used in the study were inbred mice with highly homogeneous gene backgrounds, which were maintained under well-controlled living environments. Thus, the animal subjects do not account for the large variability of human diversity regarding gene background, lifestyles (such as diet and physical activities), and underlying health conditions (like Parkinson's disease and obesity). Despite these limitations, the large quantities with detailed labels of animal datasets like ours provide sequential images covering the entire disease development and progression time course, which can be useful to test the feasibility and performance of new diagnosis and prognosis tools prior to human trials.

Despite of being a limitation, the use of preclinical models is well justified and necessary in human cancer research. As elaborated in the "Guidelines for the welfare and use of animals in cancer research" (42), animal experiments remain essential to understand the fundamental mechanisms driving malignancy and to develop individualized molecularly targeted cancer therapies for humans. With the explosion of newly created genetic animal models, the various genes, signaling pathways, and risk factors that give rise to cancer dependencies and vulnerabilities have been identified, and the responses to cancer treatments can be comprehensively tested. Thus animal models and testing not only extend our genetic, molecular, and holistic understanding of cancer but also gather necessary

safety and efficacy information that is required to introduce new drugs and diagnostic tools into clinics. Specifically, our dataset was collected from mice implanted with triple-negative breast cancer, which is the most aggressive among all subtypes of breast cancer with limited treatment options. This dataset takes advantage of our animal models, which mimics the full course of the disease development from microscopic metastasis lesions to macroscopic metastasis (whole bone failure). Currently, we have collected 2D and 3D images of the whole bone and cancer-affected regions at five-time points, along with the biometrics of the animal subjects such as their age and body weight. The Dataset is designed to expand and include additional biochemical data such as cancer burden, sera biomarkers, gene transcripts, and histological results. Preclinical research in the cancer field as well as our own studies support the similarities between the mice and humans in terms of the fundamental mechanisms driving cancer spread and growth in the bone environment. Our specific goal is to reveal specific imaging biomarkers associated with cancer growth and bone metastasis, and further make it possible to translate into clinical applications, enabling better understanding, detection, and treatment of bone metastases in human breast cancer.

