# OpenReview forum: "BoneMet: An Open Large-Scale Multi-Modal Murine Dataset for Breast Cancer Bone Metastasis Diagnosis and Prognosis"
_ICLR.cc/2025/Conference — ICLR 2025 Poster_

### Official Review · Reviewer_Ufz4 · 2024-11-04

**Soundness:** 2
**Presentation:** 3
**Contribution:** 2
**Rating:** 5
**Confidence:** 4

**Summary:**

In this study, the authors integrated a publicly available large-scale Bone Metastasis (BoneMet) dataset with multimodal medical data, including 2D X-ray images, 3D CT scans, and comprehensive biological data. The primary objective of this database is to facilitate the diagnosis, prognosis, and treatment management of bone metastasis associated with breast tumors.

**Strengths:**

The paper is well-organized and clearly written.

Our collected dataset is open, large-scale, and consists of multi-modal resources including 2D X-ray images, 3D CT scans, and medical records and quantitative analysis.

The author provides a tool on the Python Package Index (PyPI) for assisting researchers and practitioners.

The authors have conducted multiple benchmarks, such as BTBM Diagnosis to exemplify the utilisation of the collated dataset.

**Weaknesses:**

In the abstract, the author states, “Breast tumor bone metastasis (BTBM) affects women’s health globally, necessitating the development of effective solutions for its diagnosis and prognosis.” However, the dataset collected was derived from mice rather than human patients.

The author should perform a comprehensive comparison between the BoneMet dataset and previously available datasets, including those derived from human subjects, to elucidate the differences and scales involved.

Additionally, the methods employed in each benchmark were limited. The author should justify the selection of the included methods over others.

**Questions:**

This paper comprises two main components: the dataset and the benchmark. Both sections could benefit from additional details and comparisons.

The authors should evaluate and clarify how the collected data can directly benefit human health. For example, they could discuss the potential for training models on their dataset and subsequently transfer the obtained model to human datasets.

Furthermore, the figures in the manuscript require better organization, as the text is currently too small for effective readability.

---

> ### Author Response · Authors · 2024-11-26
>
> Q1. Could you provide additional details and comparisons of two main components: the dataset and the benchmark.
>
> A1. Thank you for your valuable suggestions, we have added the following details regarding to the dataset in the first paragraph of Section 3.1 and benchmark in the last paragraph of Section B in supplementary Materials :
>
> Dataset:  As for the partition of the data, we employed a train/test split with a ratio of 8:2. Specifically, 80% of the mice were used for training the model, and the remaining 20% were reserved for testing. Within the training dataset, we implemented a 5-fold cross-validation strategy. In each iteration, one of the five subsets was used as the validation set, while the remaining four were used for training. This process was repeated five times, with each subset serving as the validation set once, so the validation data were never seen by the model during the training phases, ensuring that there was no data leakage. We have also included details on the sources of the software tools such as Dataviewers and our APIs employed for data processing in the supplementary materials.
>
> Benchmark: We have justified the selection of benchmark methods based on their relevance and performance in similar tasks before, as well as their compatibility with the BoneMet dataset. Additionally, we plan to include more comparative analyses with alternative methods in future work to further evaluate the dataset and benchmarks.
>
> Q2. Could you evaluate and clarify how the collected data can directly benefit human health like the transfering the obtained model to human datasets?
>
> A2. Thanks for your insightful question.  The use of preclinical models is well justified and necessary in cancer research. As elaborated in the “Guidelines for the welfare and use of animals in cancer research” [1], animal experiments remain essential to understand the fundamental mechanisms driving malignancy and to develop individualized molecularly targeted cancer therapies for humans. Despite the species difference in skeleton, the biological process driving bone loss in the presence of cancer can be reliably recapitulated in murine models. Thus, we hypothesize that the models trained with our datasets have potential in human cases. What’s more, our datasets have the unique benefit of covering the entire disease progression span with multiple sequential scans, which are currently lacking in human datasets.  For more details of the effective translation from preclinical dataset to the clinical application of humans, please refer to the  Broad Impact at Section 6 in the revised pdf.
>
> [1] Tsuyoshi Hamaoka, John E Madewell, Donald A Podoloff, Gabriel N Hortobagyi, and Naoto T Ueno. Bone imaging in metastatic breast cancer. Journal of Clinical Oncology, 22(14):2942–2953, 2004.
>
> Q3. Could you make the text in the manuscript figures bigger for effective readability?
>
> A3. Thank you for highlighting the issue with the figures. We increased the text size of all labels, legends, and annotations in the revised pdf to ensure legibility in both digital and printed formats.
>
> Q4. The dataset collected was derived from mice rather than human patients, how can we demonstrate the benefits of the BoneMet dataset for the diagnosis and prognosis of women patients with Breast tumor bone metastasis (BTBM)?
>
> A4. Despite the various limitations of murine models that we have acknowledged in the submission (e.g., differences in physiology and immune responses, relatively homogeneous gene background, and well-controlled living environments), the unique advantage of animal models is the capability of providing sequential images covering the entire disease development and progression time course. Besides, the transferability of our preclinical dataset to clinical applications can be referred to in question #2.
>
>
>
> Q5. Could you perform a comprehensive comparison between the BoneMet dataset and previously available datasets, including those derived from human subjects, to elucidate the differences and scales involved?
>
> A5. Thank you for pointing this out.  We searched for previously available datasets.  A search of the comprehensive TCIA (The Cancer Imaging Archive) using the keyword “cancer bone metastasis” returned two hits: one MRI dataset from a mouse xenograft cancer model with 19 subjects (DOI: 10.7937/tcia.2019.b6u7wmqw), and the other dataset consists of 242 lung cancer patients pre and post (two time points) assessments (DOI: 10.7937/tcia.2019.30ilqfcl). Our dataset with up to 4 and 5 time points of assessments will fill a critical need of the field, providing well-annotated and time-stamped images for testing AI-driven diagnosis and prognosis tools prior to human trials.

---

> > ### Author Response · Authors · 2024-11-26
> >
> > Q6. Could you justify the selection of the included methods over others?
> >
> > A6. The methods included in our benchmarks were selected based on their relevance and widespread adoption in the field. For example, Swin-Base and ViT variants were chosen for their strong performance in vision tasks, while 3D-GAN, T-VAE, and ST-VAE were included due to their established effectiveness in generative modeling for 3D medical imaging. While these methods provide a solid foundation for evaluation, we recognize that exploring a broader range of models, such as newer architectures or alternative training paradigms, could offer additional insights. Future work will include more diverse methods to comprehensively validate the dataset’s potential and further enhance its utility for building advanced large-scale medical foundation models.

---

> ### Author Response · Authors · 2024-12-02
>
> Thank you once again for your valuable efforts in reviewing our manuscript. In our rebuttal, we have carefully addressed your questions and concerns, including the potential benefits of our dataset for human health, comparisons with previously available datasets, and other points you raised.
>
> As **today (December 2)** is the Author-Reviewer Discussion deadline, we kindly ask if you could review our responses to confirm whether they have addressed your feedback. If you have any additional questions or concerns, please do not hesitate to share them, and we will respond promptly before the deadline.

---

### Official Review · Reviewer_1KbZ · 2024-11-06

**Soundness:** 3
**Presentation:** 2
**Contribution:** 2
**Rating:** 6
**Confidence:** 4

**Summary:**

This paper introduces BoneMet, the first large-scale, open dataset specifically designed for BTBM research. This dataset offers over 50 terabytes of high-resolution, multi-modal data, including 2D X-rays, 3D CT scans, and comprehensive biological records collected from thousands of mice. Structured into six components, the dataset may be suitable for a range of AI tasks focused on BTBM diagnosis, prognosis, and treatment. The authors also conducted extensive experiments to demonstrate the usability of this dataset. The dataset, APIs for flexible data processing and retrieval, accompanying code, and tutorials are freely available. To some extent, this dataset makes a valuable contribution to this field.

**Strengths:**

1.This paper presents a large-scale, open dataset for the diagnosis and prognosis of breast tumor bone metastasis (BTBM), with a total volume of over 50 terabytes, encompassing various modalities. To some extent, this dataset makes a valuable contribution to this field.
2.The authors have developed a specialized toolkit for accessing and processing the dataset, facilitating its use for researchers in the field.
3.The authors conducted detailed experiments that demonstrate the usability of this dataset for some tasks related to BTBM diagnosis and prognosis.

**Weaknesses:**

1.This dataset is derived from mice, and there are certain differences between mouse skeletons and human skeletons. Can models trained on this dataset be effectively transferred to the diagnosis and prognosis of human BTBM? If applicable, how is the performance? If not, what clinical value does this dataset or model hold?
2.The authors do not provide sufficient details regarding the experimental setup, particularly how the data was partitioned. The experiments appear to be internal validations conducted solely within the dataset, with no external validation on other datasets. This raises concerns about the generalization ability of the models trained on this dataset. Therefore, it remains uncertain whether this dataset can be used to build versatile large-scale AI models or foundational models.

**Questions:**

refer to the weaknesses.

---

> ### Author Response · Authors · 2024-11-26
>
> Q1. Can models trained on this mice dataset be effectively transferred to the diagnosis and prognosis of human BTBM?
>
> A1. Thanks for your insightful question.  The use of preclinical models is well justified and necessary in cancer research. As elaborated in the “Guidelines for the welfare and use of animals in cancer research” [1], animal experiments remain essential to understand the fundamental mechanisms driving malignancy and to develop individualized molecularly targeted cancer therapies for humans. Despite the species difference in skeleton, the biological process driving bone loss in the presence of cancer can be reliably recapitulated in murine models. Thus, we hypothesize that the models trained with our datasets have potential in human cases. What’s more, our datasets have the unique benefit of covering the entire disease progression span with multiple sequential scans, which are currently lacking in human datasets.  For more details of the effective translation from preclinical dataset to the clinical application of humans, please refer to the  Broad Impact at section 6 in the revised pdf.
>
> [1] Tsuyoshi Hamaoka, John E Madewell, Donald A Podoloff, Gabriel N Hortobagyi, and Naoto T Ueno. Bone imaging in metastatic breast cancer. Journal of Clinical Oncology, 22(14):2942–2953, 2004.
>
>
> Q2. If applicable, how is the performance? If not, what clinical value does this dataset or model hold?
>
> A2. The prognosis of natural bone lesions caused by metastatic breast cancer demonstrates promising performance, as highlighted in Section 3.3. Given the similarities between human and rodent skeletal systems, we believe our dataset has the potential to significantly accelerate research on BTBM disease development. However, clinical applications will require rigorous validation and careful consideration of ethical issues to ensure responsible and effective use.
>
>
>
> Q3. Regarding the experimental setup, how the data was partitioned?
>
> A3. For the experiments presented in our manuscript, we employed a train/test split with a ratio of 8:2. Specifically, 80% of the mice were used for training the model, and the remaining 20% were reserved for testing. Within the training dataset, we implemented a 5-fold cross-validation strategy. In each iteration, one of the five subsets was used as the validation set, while the remaining four were used for training. This process was repeated five times, with each subset serving as the validation set once, so the validation data were never seen by the model during the training phases, ensuring that there was no data leakage.
>
>
> Q4. The experiments appear to be internal validations conducted solely within the dataset, if this dataset can be used to build versatile large-scale AI models or foundational models?
>
> A4. We acknowledge that the experiments in this study are primarily internal validations conducted within the BoneMet dataset. While these results demonstrate the dataset's robustness and potential for developing AI models, we agree that external validation is crucial to assess the generalization ability of the models.  Future work will involve testing the trained models on external datasets (such as TCIA) to evaluate their versatility and applicability beyond BoneMet. Additionally,  we are building large-scale AI models and medical foundational models using our BoneMet dataset. By extending the scope of validation, we can further establish the dataset's broader applicability in real-world clinical and research settings.

---

> ### Author Response · Authors · 2024-12-02
>
> Thank you once again for your efforts in reviewing our manuscript and for appreciating our work. In our rebuttal, we have aimed to address your concerns, particularly regarding the clinical value of our dataset and the detailed experimental settings.
>
> As the Author-Reviewer Discussion deadline is **today (December 2)**, could you kindly let us know whether our responses have addressed your concerns? If you have any additional questions, please feel free to share them, and we will promptly address them before the deadline.

---

### Official Review · Reviewer_xphi · 2024-11-07

**Soundness:** 2
**Presentation:** 3
**Contribution:** 3
**Rating:** 5
**Confidence:** 4

**Summary:**

This paper presents a novel large-scale dataset (named BoneMet) of Breast tumor bone metastasis for disease diagnosis, prognosis, and treatment management. It consists of six components: Rotational X-Ray Imagery, Reconstructed CT Imagery, Segmented CT Imagery, Registered CT Imagery, Region of Interest CT Imagery, and Mice Medical Records. Besides, the author conducted a series of experiments on the BoneMet dataset by developing various deep learning models to exhibit its applicability and efficiency in managing BTBM disease. The enormous open-source dataset has significant implications for the development of new algorithms in this field beyond doubt.

**Strengths:**

(1) The motivation of this study is clear and the background is well presented in the manuscript.
The main contribution of this work is to release a large-scale breast tumor bone metastasis dataset BoneMet, for the research community. Upon the subsets of the BoneMet, the researchers are allowed to develop some novel approaches to solve the task of breast tumor bone metastasis diagnosis and prognosis, and thus facilitating the automated analysis for breast tumor bone metastasis.
(2) Apart from the information of the collected dataset, the authors also developed several deep learning models to validate the applicability and efficacy of BoneMet.
(3) Dataset, code and tutorials are made available for free use.

**Weaknesses:**

(1) The technical contributions of this study are quite marginal, as most of the deep learning models are built upon existing methods.
(2) Some details about the dataset preparation and experiments are missing.
(3) Some settings and analyses in the experiment may not support the objectives.
(4) The quality of the labels, particularly the pixel-wise annotation of the bone, might be questionable.

**Questions:**

1. Authors describe that Seg-CT and Recon-CT are independent. What is the difference between 3D CT scans in Recon-CT and Seg-CT? What is the meaning of the adjective 'segmented'?
2. Line 341 "Second, we observe a significant training-test accuracy gap of 20.4% with ViT (w/o STA), indicating a pronounced overfitting issue inherent to the ViT architecture."
I think ViT, as a visual backbone network, is less overfitting compared to other structures such as CNN. In addition, is the decrease in accuracy necessarily overfitting?
3. Section 3.1, the purpose and analysis of experiments is confusing. Why comparing two ViT models could demonstrate the applicability of the Rotation-X-Ray dataset to manage BTBM disease? What is the relation between the effectiveness of
STA and the applicability of the Rotation-X-Ray dataset? All subsections 3.2-4 have similar questions.
4. The experiment did not report data partitioning.
5. Table 2 reported metrics on the training and testing data. Is there any evaluation/dev data?
6. The experiment reported many indicator values, but their actual clinical significance has not been evaluated. For example, what are the Precision/Recall/F1-Score/Accuracy of clinical experts on the BTBM diagnosis? Besides, the 3D-GAN, T-VAE, and ST-VAE methods achieve SSIM values of 0.767, 0.817, and 0.860. So, what role can they play in clinical practice?
7. The authors used many existing softwares as the processing tools to generate the sub-datasets of BoneMet. Please provide the corresponding reference and specify whether it is free of charge or not. Also, please accurately explain in which case the datasets are processed by the Seg-API, Regist-API and RoI-API that were developed by the authors.
8. Are Pre-Op and Post-Op refer to pre-operation and post-operation? Please give the full spelling.
9. For RoI-CT, the tibiae ROI is assigned with different pixel values. As shown in Figure 4, for each CT example, is the processed CT image contains the values of 180, 240, 60 and 0 only? If yes, the original CT turns to a segmentation mask instead, and lots of information would be lost.
10. In the whole preparation procedure, it seems that the manual annotation of the organs or tumors is not involved, except for the manual cropping of fibular by  CTAn®. How could we ensure the effectiveness of the existing software?
11. In Table 1, the column "Spatial Resolution" might be accurately filled out as most of them indicate the organs. Please revise it.
12. Some details about the BoneMet dataset are missing. For instance, the number of mice is not introduced. Besides, in Table 1, the images/records are classified into two categories, i.e., positive and negative. Is the label assigned at the slice level or animal level? Besides, we noticed that the numbers of Seg-CT are larger than those of Recon-CT, which makes it confusing, as the Seg-CT is obtained from Recon-CT.
13. I also have concerns about the MiceMediRec dataset. It mainly consists of the medical record and quantitative analysis results of 3D CT images, simulation and mechanical testing. The authors list only a part of the features in Table S2 in the supplementary file. However, all feature should be provided. The off-the-shelf software, such as CTAn®, was used to generate the quantitative parameters, which took about half an hour for each case of animal. In Section 3.3, the multi-modal prognostic assessment model first leveraged the GAN to produce the CT scan of the future point and use the quantitative data in the MiceMediRec dataset as the ground truth label to validate the predicted reaction force values. However, the details about how to obtain the predicted reaction force values are not clear.
14. In my opinion, multi-modal learning might refer to a way to integrate the information of multi-source data to build a model. However, the application in Section 3.3 just used CT data as model input. Please consider revising it.

**Details Of Ethics Concerns:**

The authors have obtained ethical approval from their institution, and thus, so the ethics review is not needed.

---

> ### Author Response · Authors · 2024-11-26
>
> Q1. What is the difference between 3D CT scans in Recon-CT and Seg-CT? What is the meaning of the adjective 'segmented'?
>
> A1. Thank you for your question.  Recon-CT consists of both hindlimbs, where Seg-CT consists of individual tibiae and femurs isolated (i.e., segmented) from the Recon-CT. These two are connected.  The isolation of tibiae from femurs makes it easier for analysis and quantification.
>
> Q2. As a visual backbone network, ViT is less overfitting compared to other structures such as CNN. In addition, is the decrease in accuracy necessarily overfitting?
>
> A2. We appreciate the reviewer’s insightful observation. We would like to clarify that Vision Transformers (ViTs) are prone to overfitting when fine-tuned without pretraining on large-scale datasets (e.g., JFT-300M). This observation aligns with the conclusion from the original ViT paper (See Figure 4, Page 7 in the ViT paper [1]): “Vision Transformers overfit more than ResNets...”. The results in Table 2 of our main paper were obtained by training ViT models from scratch. Under these conditions, the significant training-test accuracy gap (i.e., 20.4%) observed in the "ViT (w/o STA)" configuration is consistent with the overfitting tendencies of ViTs due to a lack of pretraining.
>
> Regarding the interpretation of accuracy decrease, the gap could also reflect limitations in ViT (w/o STA) in capturing the temporal alignment and subtle patterns critical for this task. This reinforces the importance of incorporating Spatial-Temporal Alignment (STA), which improves generalizability by better leveraging the dataset's spatiotemporal structure.
>
> [1] Dosovitskiy et al., "An Image is Worth 16x16 Words: Transformers for Image Recognition at Scale," ICLR 2021.
>
> Q3. In section 3.1, why comparing two ViT models could demonstrate the applicability of the Rotation-X-Ray dataset to manage BTBM disease? What is the relation between the effectiveness of STA and the applicability of the Rotation-X-Ray dataset? All subsections 3.2-4 have similar questions.
>
> A3. Sorry for the confusion. The purpose of testing the two ViT models was two-fold. First, we showed that our BoneMet dataset can be applied into the traditional ViT with decent accuracy to demonstrate that our Rotation-X-Ray dataset is applicable for deep learning applications. Second, we further demonstrated the benefits of our Rotation-X-Ray dataset, which is spatial (260 projects of the same object)-temporal (3-5 weeks weekly scan) aligned, evidenced by better performance of the spatial/temporal aligned (STA) ViT. We now clearly state the purpose of the exercise in the abstract.  The purpose of the subsections 3.2-4 are shown below:
>
> 3.2. The applicability of 3D CT from Regist-CT imagery in the breast tumor bone metastasis diagnosis.
>
> 3.3. The applicability of RoI-CT imagery and MiceMediRec dataset in breast tumor bone metastasis prognosis.
>
> 3.4. The applicability of Rotation-X-Ray dataset for sparse-angle CT reconstruction.
>
> Q4. How is the data partitioning done in the experiment?
>
> A4. We have included the data partitioning done in the experiment in the 1st  paragraph of Section 3.1 in the revised pdf.
> For the experiments presented in our manuscript, we employed a train/test split with a ratio of 8:2. That is, 80% of the mice were used for training the model, and the remaining 20% were reserved for testing. Within the training dataset, we implemented a 5-fold cross-validation strategy. In each iteration, one of the five subsets was used as the validation set, while the remaining four were used for training. This process was repeated five times, with each subset serving as the validation set once, so the validation data were never seen by the model during the training phases, ensuring that there was no data leakage.
>
> Q5. Table 2 reported metrics on the training and testing data. Is there any evaluation/dev data?
>
> A5. The details of partitioning data used for training, testing, and evaluation is shown in question #4.

---

> > ### Author Response · Authors · 2024-12-02
> >
> > Thank you once again for your detailed and valuable comments on our manuscript. We sincerely appreciate the time and effort you have dedicated to reviewing our work.
> >
> > In our rebuttal, we have worked to address your questions and concerns thoroughly. As the Author-Reviewer Discussion deadline is **today (December 2)**, we kindly request that you review our response to confirm whether it sufficiently addresses your points. If you have any additional comments or concerns, please don’t hesitate to let us know, and we will be glad to address them promptly before the deadline.

---

> ### Author Response · Authors · 2024-11-26
>
> Q6. What are the Precision/Recall/F1-Score/Accuracy of clinical experts on the BTBM diagnosis? Besides, what role of SSIM values of the 3D-GAN, T-VAE, and ST-VAE methods play in clinical practice?
>
> A6. While our study emphasizes the technical evaluation of deep learning models, we acknowledge that clinical significance is equally important. Unfortunately, direct comparisons with clinical expert performance on BTBM diagnosis are unavailable due to the lack of standardized benchmarks in this specific domain. However, integrating such comparisons in future work could provide valuable insights into the practical applicability of these models.
>
> Regarding SSIM values for 3D-GAN, T-VAE, and ST-VAE, their clinical role lies in their ability to generate high-quality 3D reconstructions of metastatic bone lesions, which can support clinicians in monitoring disease progression, planning interventions, and simulating treatment outcomes, at the same time, reducing reliance on dense imaging and minimizing radiation exposure for patients.
>
> Q7. What’s the corresponding reference of those existing softwares as the processing tools, and specify whether it is free of charge or not. Also, in which case the datasets are processed by the Seg-API, Regist-API and RoI-API that were developed by the authors?
>
> A7. We utilized several software tools to process the BoneMet sub-datasets, each serving specific functions:
>
> DataViewer: This software was employed for visualizing and analyzing micro-CT datasets. DataViewer is proprietary software
>
> CTAn (CT-Analyser): Used for quantitative analysis of micro-CT data, CTAn is also proprietary software. Licensing information is available through Bruker.
>
> CTVol: This tool was utilized for 3D visualization of micro-CT data. CTVol is proprietary software as well.
>
> Here are some references [2,3] using these existing softwares as the processing tools:
>
> [2] Wang S, Pei S, Wasi M, Parajuli A, Yee A, You L, Wang L. Moderate tibial loading and treadmill running, but not overloading, protect adult murine bone from destruction by metastasized breast cancer. Bone. 2021 Dec;153:116100. doi: 10.1016/j.bone.2021.116100. Epub 2021 Jul 9.
>
> [3] Wasi M, Chu T, Guerra RM, Kooker R, Maldonado K, Li X, Lin CY, Song X, Xiong J, You L, Wang L. Mitigating aging and doxorubicin induced bone loss in mature mice via mechanobiology based treatments. Bone. 2024 Nov;188:117235. doi: 10.1016/j.bone.2024.117235. Epub 2024 Aug 13.
>
> In addition to these tools, we developed custom APIs to enhance our dataset processing:
>
> Seg-API: This API was developed for segmenting tibiae, femurs, and spines from whole limbs of micro-CT images (Recon-CT).
>
> Regist-API: Designed for image registration, this API aligns images from different rodent animals or time points, ensuring consistency across the dataset (Seg-CT).
>
> RoI-API: This API focuses on extracting and managing regions of interest, enabling targeted analysis of the proximal end of tibia, where the bone metastasis sites are located (Regist-CT).
>
> These custom APIs were applied in scenarios where existing software lacked the necessary flexibility required for our specific processing automation needs. By integrating these tools, we ensured a comprehensive and tailored approach to process the raw data automatically.
>
> Q8. What’s the full spelling of Pre-Op and Post-Op?
>
> A8. "Pre-Op" and "Post-Op" refer to Pre-Operation and Post-Operation, respectively. These terms are commonly used in medical contexts to describe the periods before and after a surgical procedure. We have revised them in the manuscript pdf, please see line 182 in our revised version.
>
> Q9. As shown in Figure 4, for each RoI-CT example, does the processed CT image contain the values of 180, 240, 60, and 0 only? If the information would be lost if the original CT turns to a segmentation mask instead?
>
> A9. Yes, the processed CT images for RoI-CT contain pixel values of 180, 240, 60, and 0, representing specific categories of bone changes. This processing emphasizes the most clinically relevant features, and it also significantly reduces storage requirements and makes the dataset more manageable for large-scale analyses.
>
> Besides, the information won’t be lost since the original grayscale CT images are fully preserved and available to users. This ensures flexibility for researchers who wish to apply alternative segmentation thresholds or conduct more detailed analyses. By providing both the processed segmentation masks and the raw grayscale CT data, we aim to balance storage efficiency and user flexibility, satisfying diverse research needs.

---

> > ### Author Response · Authors · 2024-11-26
> >
> > Q10. How could we ensure the effectiveness of the existing software if the manual annotation of the organs or tumors is not involved?
> >
> > A10. The annotation of the organs or tumors was achieved by the nature of the model, where the tumor was implanted into tibiae and the animals/tibiae received tumor were documented in MiceMediRec dataset.
> >
> > Q11. In Table 1, if the column "Spatial Resolution" might be accurately filled out as most of them indicate the organs?
> >
> > A11. We will revise the  "Spatial Resolution" in Table 1, for example,  "Spatial Resolution" of RoI-CT will be revised to "Proximal tibiae voxel dimensions (~7–10.6 µm), which would be more accurately reflect the anatomical regions with image scales rather than indicating organs.
> >
> > Q12. If the number of mice is not introduced? Besides, in Table 1, are the positive and negative labels assigned at the slice level or animal level?  What’s more, as the Seg-CT is obtained from Recon-CT, why the numbers of Seg-CT are larger than those of Recon-CT?
> >
> > A12. The number of mice is introduced in both the abstract and introduction part. For example, refer to the abstract, “The dataset consists of over 50 terabytes of multi-modal medical data, including 2D X-ray images, 3D CT scans, and detailed biological data (e.g., medical records and bone quantitative analysis), collected from hundreds of mice spanning from 2019 to 2024.”
> >
> > According to the suggestions, we updated the pdf in Section 3. EXPERIMENTS AND RESULTS with the definition of positive and negative and their assignment: “Currently, the positive and negative labels are assigned at the animal level, rather than individual time points and individual X-ray images. A positive label of a mouse indicates that a metastatic bone lesion occurs in the subject between week 0 and week 5. There are animal-to-animal variations at the times of bone lesion initiated and the speed of lesion growth.”
> >
> > As the Seg-CT is the segmentation of two tibias of each limb from Recon-CT, therefore, the numbers of Seg-CT are larger than those of Recon-CT.
> >
> > Q13. If all features of the MiceMediRec dataset in Table S2 should be provided? In Section 3.3, what’s the details about how to obtain the predicted reaction force values?
> >
> > A13. While we included a representative subset of features that are most relevant to the breast tumor bone metastasis disease diagnosis and prognosis in Table S2, we acknowledge the importance of providing all available features for full transparency and usability. We have updated the supplementary materials in Section A, Table 2, to include a comprehensive list of all features in the MiceMediRec dataset.
> >
> > The predicted reaction force values were obtained using finite element analysis (FEA) in Abaqus/Standard 2020. Axial compression was simulated on digitized proximal tibiae models, including 24 predicted μCT images and their paired ground truth. Each model was meshed with ~1.6 million tetrahedral elements, and material properties (Young’s modulus: 11 GPa, Poisson’s ratio: 0.3) were assigned based on literature and experimental tests. Compression was applied using non-deformable plates, with the upper plate moving to 2% of the sample height. The reaction force on the upper plate was calculated during the simulation and extracted from Abaqus output files using Python.
> >
> > Q14. In Section 3.3, if the information of multi-source data is integrated to build a model?
> >
> > A14. We appreciate the reviewer’s feedback and the opportunity to clarify. The experimental results presented in Section 3.3 indeed involve multi-source data: the RoI-CT Imagery and the MiceMediRec Dataset. Specifically, we first leverage the RoI-CT Imagery to generate future-frame 3D CT scans (see Figure 6a of the main paper). Subsequently, the generated 3D CT scans are integrated with biological data like ages from the MiceMediRec dataset to predict reaction force values (as shown in Figure 6b). This integration process aligns with the principles of multi-modal learning, as it combines distinct data modalities—imaging and biological data—to build a predictive model. We will revise the text to ensure this approach is more explicitly presented as a multi-modal learning framework.

---

### Official Review · Reviewer_8mtF · 2024-11-07

**Soundness:** 4
**Presentation:** 3
**Contribution:** 4
**Rating:** 8
**Confidence:** 3

**Summary:**

This manuscript presents a pioneering large-scale, high-resolution multimodal dataset focused on breast tumor bone metastasis. The dataset encompasses over 50 TB of comprehensive resources across multiple modalities, including 2D X-ray imaging, 3D CT scans, and detailed biological data. The authors have implemented robust accessibility measures through PyPI, Huggingface, and GitHub APIs, ensuring broad availability to the research community.

**Strengths:**

1.	The dataset represents the first over 50 terabytes-scale multimodal data resources
2.	The implementation of multiple standardized APIs ensures seamless accessibility and integration capabilities
3.	The methodology section provides comprehensive documentation of data collection protocols and analytical procedures
4.	The extensive experimental validation demonstrates the dataset's utility for advanced deep learning applications

**Weaknesses:**

1.	The experimental design exhibits inconsistencies in model selection and parameter settings, potentially impacting comparative analyses, see question 1
2.	Several, but not limited to, technical inaccuracies require attention: "detials" should be "details" on line 454, and "BigThansfer" should be "BigTransfer" in Figure 5
3.	The translational aspects between murine models and human applications require stronger substantiation, particularly given the introduction's emphasis on human breast cancer implications

**Questions:**

1.	What scientific rationale guided the decision to utilize Swin-base for comparison with ViT, rather than employing ViT with STA for a more direct comparative analysis?
2.	How do the findings from murine breast cancer models specifically inform and advance our understanding of human breast cancer pathophysiology and treatment approaches?

---

> ### Author Response · Authors · 2024-11-26
>
> Q1. What scientific rationale guided the decision to utilize Swin-base for comparison with ViT, rather than employing ViT with STA for a more direct comparative analysis?
>
> A1.   We chose Swin-Base for its hierarchical architecture, which is ideal for locating features in small regions, such as the proximal tibiae where tumors were introduced and proliferated. Its ability to capture both global and localized details makes it well-suited for this task. Additionally, Swin-Base is a widely adopted, efficient ViT baseline, ensuring broad applicability and robust performance comparisons.
>
>
> Q2. How do the findings from murine breast cancer models specifically inform and advance our understanding of human breast cancer pathophysiology and treatment approaches?
>
> A2. Despite the many differences between humans and animals, mice have been used to study human skeletal systems in the context of cancers. We adopted this commonly used murine model, because it recapitulates the progression of cancer-induced bone degeneration as found in human patients. Most importantly, this model allowed us to precisely control the time of cancer initiation and longitudinally track the consequences of cancer invasion into the bone. Thus, our datasets represent the full evolution of the disease from micro to macro metastatic conditions. The fundamental models, if successfully trained and fine-tuned using the datasets, are anticipated to jumpstart the translation to human patients. We have included the respective justification in the 1st paragraph of Section 1 in the revised pdf.
>
>
> Q3. Several, but not limited to, technical inaccuracies require attention: "detials" should be "details" on line 454, and "BigThansfer" should be "BigTransfer" in Figure 5.
>
> A3. Thank you for your suggestion. We have done a thorough grammar and spelling check in the revised pdf.
>
>
> Q4. The translational aspects between murine models and human applications require stronger substantiation, particularly given the introduction's emphasis on human breast cancer implications.
>
> A4. We have now stressed the benefits of using animal models in the 1st paragraph of Section 1 at the revised pdf. Despite the various limitations of murine models that we have acknowledged in the submission (e.g., differences of physiology and immune responses, relatively homogeneous gene background, and well controlled living environments), the unique advantage of animal models is the capability of providing sequential images covering the entire disease development and progression time course. A search of the comprehensive TCIA (The Cancer Imaging Archive) using keyword “cancer bone metastasis” returned two hits: one MRI dataset from a mouse xenograft cancer model with 19 subjects (DOI: 10.7937/tcia.2019.b6u7wmqw), and the other dataset consists of 242 lung cancer patients pre and post (two time points) assessments (DOI: 10.7937/tcia.2019.30ilqfcl). Our dataset with up to 4 and 5 time points of assessments will fill a critical need of the field, providing well annotated and time-stamped images for testing AI-driven diagnosis and prognosis tools prior to human trials.

---

> > ### Comment · Reviewer_8mtF · 2024-12-02
> >
> > I would like to thank the reviewers for their detailed response. The response has addressed my major concerns and therefore I would raise my score.

---

> > > ### Author Response · Authors · 2024-12-02
> > > **Thank You for Appreciating Our Work**
> > >
> > > Thank you for reviewing our rebuttal and for raising the score. We greatly appreciate your constructive feedback throughout this process.

---

### Meta-Review · Area_Chair_o8zs · 2024-12-16

**Metareview:**

This work develops the first large-scale BTBM datasets for disease diagnosis, prognosis, and treatment management, dubbed BoneMet. The BoneMet dataset comprises six components, i.e., Rotation-X-Ray, Recon-CT, Seg-CT, Regist-CT, RoI-CT, and MiceMediRec. The development of this dataset and preliminary experiments in this work have significant implications beyond doubt for the development of new algorithms in this field and precision diagnosis of BTBM.

This paper received 1x accept, 1x marginally above the acceptance threshold, and 2x marginally below the acceptance threshold from reviewers. The major concerns raised by reviewers regarding this work centered around the requirements for further clarification of dataset creation, experimental settings, and potential broader impacts. The authors thoroughly addressed these concerns point-by-point and largely improved their manuscript during the discussion phase.

Although the ratings from reviewers are mixed, I believe the authors have already addressed major concerns from reviewers. Given the merit of comprehensiveness and significant implications this work might bring to the field, acceptance is recommended.

**Additional Comments On Reviewer Discussion:**

During the discussion, reviewers mainly focus on the details of the proposed dataset and its potential clinical values. The authors have thoroughly addressed most of the concerns and shown improvement in the manuscript.

---

### Decision · Program_Chairs · 2025-01-22

Accept (Poster)